# Antiviral Activity of Benzavir-2 against Emerging Flaviviruses

**DOI:** 10.3390/v12030351

**Published:** 2020-03-22

**Authors:** Yong-Dae Gwon, Mårten Strand, Richard Lindqvist, Emma Nilsson, Michael Saleeb, Mikael Elofsson, Anna K. Överby, Magnus Evander

**Affiliations:** 1Department of Clinical Microbiology, Virology, Umeå University, 90185 Umeå, Sweden; kwon.yongdae@umu.se (Y.-D.G.); marten.strand@umu.se (M.S.); richard.lindqvist@umu.se (R.L.); emma.nilsson@umu.se (E.N.); anna.overby@umu.se (A.K.Ö.); 2Umeå Centre for Microbial Research (UCMR), Umeå University, 90187 Umeå, Sweden; mikael.elofsson@umu.se; 3The Laboratory for Molecular Infection Medicine Sweden (MIMS), Umeå University, 90187 Umeå, Sweden; 4Department of Chemistry, Umeå University, 90187 Umeå, Sweden; michael.saleeb@umu.se

**Keywords:** benzavir-2, flavivirus, Zika virus, antiviral drugs

## Abstract

Most flaviviruses are arthropod-borne viruses, transmitted by either ticks or mosquitoes, and cause morbidity and mortality worldwide. They are endemic in many countries and have recently emerged in new regions, such as the Zika virus (ZIKV) in South-and Central America, the West Nile virus (WNV) in North America, and the Yellow fever virus (YFV) in Brazil and many African countries, highlighting the need for preparedness. Currently, there are no antiviral drugs available to treat flavivirus infections. We have previously discovered a broad-spectrum antiviral compound, benzavir-2, with potent antiviral activity against both DNA- and RNA-viruses. Our purpose was to investigate the inhibitory activity of benzavir-2 against flaviviruses. We used a ZIKV ZsGreen-expressing vector, two lineages of wild-type ZIKV, and other medically important flaviviruses. Benzavir-2 inhibited ZIKV derived reporter gene expression with an EC_50_ value of 0.8 ± 0.1 µM. Furthermore, ZIKV plaque formation, progeny virus production, and viral RNA expression were strongly inhibited. In addition, 2.5 µM of benzavir-2 reduced infection in vitro in three to five orders of magnitude for five other flaviviruses: WNV, YFV, the tick-borne encephalitis virus, Japanese encephalitis virus, and dengue virus. In conclusion, benzavir-2 was a potent inhibitor of flavivirus infection, which supported the broad-spectrum antiviral activity of benzavir-2.

## 1. Introduction

Flaviviruses are small, enveloped, single-stranded positive-sense RNA viruses that cause serious diseases in humans and animals [1,2]. Most flaviviruses are arthropod-borne viruses transmitted to vertebrate hosts by either mosquitoes or ticks [3]. Several members of the flavivirus genus, such as the Zika virus (ZIKV), dengue virus (DENV), yellow fever virus (YFV), West Nile virus (WNV), Japanese encephalitis virus (JEV), and tick-borne encephalitis virus (TBEV), are highly pathogenic to humans [4,5,6].

The recent outbreaks of flaviviruses have raised a global concern, since several reports linked the recent ZIKV outbreak in the Americas with an increase of microcephaly cases in fetuses and newborns and an increase of Guillian–Barre syndrome cases [7,8,9]. In addition, the recent YFV outbreak in Africa caused approximately 600 deaths [10]. Hence, flaviviruses pose a global threat to the population, especially in areas that are naive to infection. Because of climate change and increasing mobility of people, flaviviruses have emerged outside endemic areas.

In order to limit flavivirus outbreaks, antiviral drugs are urgently needed. However, up to date, there are no approved antiviral therapies for flaviviruses. Several preclinical and investigational drugs have been assessed for anti-flavivirus activity [11,12,13,14]. However, flaviviruses often escape antivirals that target specific viral proteins because of the high mutation rates [15]. Given this lack of specific treatment options for flaviviruses, a host directed therapy could represent a strategy for the discovery of pan-flavivirus antiviral agents [13].

We have previously identified and optimized anti-adenoviral compounds resulting in benzavir-2 (Figure 1A), a novel antiviral compound with broad antiviral activity against both DNA viruses, human adenovirus (HAdV) [16,17], herpes simplex virus type 1 and type 2 (HSV-1, HSV-2) [18], and, recently, RNA virus, and rift valley fever virus (RVFV) [19]. Here, we explore the antiviral activity of benzavir-2 against flaviviruses that urgently need effective therapeutic interventions. We used a *Zoanthus species* green fluorescence protein (ZsGreen), expressing infectious ZIKV as a primary tool to study anti-flavivirus activity and then expanding it to investigate the effect on wild-type (wt) ZIKV and several other pathogenic flaviviruses. The antiviral activity of benzavir-2 was compared with the activity of other antiviral drugs, favipiravir, and ribavirin, which were recently reported to display antiviral activity against flaviviruses [20]. This study demonstrated potent antiviral activity of benzavir-2 against flaviviruses and thus added a new group of viruses to its broad spectrum of antiviral activity.

## 2. Materials and Methods

### 2.1. Bio-Containment Levels Statement

In this study, all of the experiments were performed in the bio-containment level 2+ laboratory (BSL2+), except the experiments for “focus-forming assay” and “antiviral assay against other flaviviruses”, which were performed in the bio-containment level 3 laboratory (BSL3).

### 2.2. Rescue of Recombinant ZIKV-Zsgreen, Titration, and Propagation

Plasmids containing a full-length infectious clone (icDNA) of the Brazilian ZIKV isolate, containing the ZsGreen reporter (pCCI-SP6-ZIKV-ZsGreen) were kindly provided by Dr. A. Merits (Tartu University, Tartu, Estonia), and the strategy previously described for ZIKV icDNA rescue was used [21]. Briefly, 5 µg of pCCI-SP6-ZIKV-ZsGreen were linearized using AgeI restriction enzyme prior to in vitro transcription. The resulting DNA fragments were purified using the NucleoSpin Gel and PCR Clean-up kit (Macherey-Nagel, Germany) and eluted with 6 µL water. The mMESSAGE mMACHINE SP6 transcription kit (Thermo Scientific, USA) was used to synthesize in vitro transcribed RNA from 3 µL of cDNA in a 20 µL reaction volume. The capped RNA transcripts were transfected into 4 × 10^5^ Vero B4 cells [22] using the TransIT^®^-mRNA Transfection Kit (Mirus Bio, Madison, WI, USA). Transfected cells were monitored daily and, when significant, the cytopathogenic effect and the ZsGreen signal was observed, the stocks of rescued viruses (rZIKV-ZsGreen), passage number 0 (p0 stocks) were harvested, centrifuged at 1500 rpm for 5 min to remove cellular debris, and aliquoted with 20% fetal bovine serum (FBS). Aliquots were stored at −80 °C until use.

The virus concentration of p0 stocks was determined by plaque assay, as previously described [23,24]. Briefly, 500 µL of tenfold serial dilutions of the virus samples were added to a 6-well plate containing a confluent monolayer of Vero B4 cells. After 1 h incubation with shaking every 15 min, the virus suspension was discarded and an overlay was added that contained 2 mL of Dulbecco’s modified Eagle medium (DMEM) with 2% FBS, 0.5% minimum essential amino acids, 0.5% sodium pyruvate, 0.5% L-Glutamine, 1% penicillin/streptomycin, and 1% carboxymethylcellulose (CMC, Sigma Aldrich, St. Louis, MO, USA). After 4–5 days of incubation, the cells were fixed with 4% paraformaldehyde and stained with 1% crystal violet solution (1% crystal violet powder, 20% *w*/*v* methanol in distilled H_2_O). Plaques were counted and viral titers were calculated as plaque-forming units (PFUs)/mL.

To propagate the p0 stock, confluent Vero B4 cells grown in 25T flasks (Sarstedt, Nümbrecht, Germany) were infected at 0.1 multiplicity of infection (MOI). The supernatants (p1 stocks) were collected when a cytopathic effect (CPE) and the ZsGreen signal were observed. The p1 stocks were titrated and used for the dose-response analysis and time of addition assay.

### 2.3. Virus and Cell

The MR-766 isolate of ZIKV (collected from a sentinel rhesus monkey in the Zika forest of Uganda in April 1947) was kindly provided by Dr. G. Dobler (Bundeswehr Institute of Microbiology, Munich, Germany). The ZIKV BeH819015 strain (GI: 975885966; isolated in Brazil in 2015) was rescued from the icDNA. The TBEV strain Hypr71 was kindly provided by Dr. G. Dobler (Bundeswehr Institute of Microbiology, Munich, Germany). The YFV (strain Asibi) was kindly provided by Dr. M. Niedrig (Robert Koch Institute, Berlin, Germany). The JEV (strain Nakayama), WNV (isolated in 2003 in Israel WNV_0304h_ISR00, passage number 5), and DENV-2 (PNG/New Guinea C) strains were kind gifts from Dr. S. Vene (Public Health Agency of Sweden, Stockholm, Sweden). Virus strains were amplified and titrated in Vero B4 cells. 

Vero B4 cells (African green monkey kidney) were maintained at 37 °C, 5% CO_2_ in Dulbecco’s modified Eagle medium (DMEM), supplemented with 5% fetal bovine serum (FBS), 2% 4-(2-hydroxyethyl)-1-piperazineethanesulfonic acid (HEPES), and 1% Penicillin/streptomycin. For virus infection, cell maintenance medium was used, containing the same components, except at a lower FBS concentration (2%).

### 2.4. Reagents

Ribavirin was purchased from Sigma Aldrich (St. Louis, MO, USA). Favipiravir was purchased from Selleckchem (Munich, Germany). Benzavir-1, benzavir-2, and compounds 17d and 35f were synthesized as previously described [17,19] and were dissolved in DMSO. The chemical structures of the compounds analyzed in this study is illustrated in Figure 1A.

### 2.5. Dose-Response Analysis

The six compounds were analyzed using a fluorescent intensity assay that quantifies the ZsGreen expression, as previously described [19]. The day before infection, 10^4^ Vero B4 cells per well were seeded in 96-well black plates with transparent bottoms (Corning, Corning, NY, USA). The next day, compounds were two-fold serially-diluted nine steps in dimethyl sulfoxide (DMSO) to final concentrations from 100 µM to 0.19 µM in 0.5 % DMSO. The compound mixture was mixed with 2000 PFUs of rZIKV-ZsGreen virus in a total volume of 200 µL. Then, 200 µL of compound/virus mixture was added to the cells. The plates were incubated for 24 h at 37 °C in a 5% CO_2_ incubator.

After incubation, the medium was removed, and cells were fixed with 4% paraformaldehyde for 1 h at room temperature and washed with phosphate-buffered saline (PBS). The fixed cells were counterstained with 300 nM 4′,6-diamidino-2-phenylindole (DAPI) in PBS. The Trophos plate runner HD^®^ (Trophos, Roche Group, Basel, Switzerland) identified all individual ZsGreen-expressing cells or DAPI-stained cells and quantified the fluorescence intensity in each well. The infection rates were calculated by dividing the number of infected cells (ZsGreen-expressing cells) to the total number of cells (DAPI-stained cells). The EC_50_ values were calculated by non-linear regression analysis with a variable slope, using GraphPad Prism 7.0 software. Furthermore, the EC_90_ values were calculated by Quick → Calcs, which is provided at GraphPad website [25]

### 2.6. Cell Toxicity Assay

The cellular toxicity of compounds was assessed by using the Resazurin cell viability assay (Sigma–Aldrich), which assesses the metabolic activity of living cells [26]. To analyze cell viability post compound treatments, 10 µL (40 µM final concentration) of resazurin was added per well and incubated for 3–4 h at 37 °C in a 5% CO_2_ incubator, and the fluorescent intensity was measured at 535 nm (emission spectra 590 nm).

### 2.7. Plaque Inhibition Assay 

Vero B4 cells (2 × 10^5^ cells per well) were seeded in 12-well plates the day prior to infection. The two wt ZIKV lineages (300 PFUs), together with 2.5 µM of benzavir-2 in DMEM with 1% CMC, was added. The cells were then incubated for 4 days. Next, the cells were fixed with neutral-buffered formalin (4%) and stained with 1% crystal violet solution (1% crystal violet powder, 20% *w*/*v* methanol in distilled H_2_O). To measure plaques, we used the open-source software solution, Plaque2.0 [27].

### 2.8. Progeny Virus Reduction Assay

Vero B4 cells were seeded in 12-well plates (2 × 10^5^ cells per well) and wt ZIKV (300 PFU) solution, together with 2.5 µM of benzavir-2 in DMEM, was added, and the cells were incubated for 4 days. The supernatants were collected from infected cells with or without compounds. Next, the collected supernatants were transferred to new Vero B4 cell layers (indicator cells) for 1 h and replaced with DMEM with 1% CMC (for CPE observation and crystal violet staining) or with fresh DMEM (for qRT-PCR), respectively. The plates were analyzed by observing virus-induced CPE 2 days post-infection, and plaque formation was determined by crystal violet staining on day 4. The supernatants were collected and further used for viral RNA isolation and quantification of virus RNA in the supernatant.

### 2.9. Real-Time (RT) PCR for the Detection of ZIKV RNA in Cell Culture Supernatant

To quantify the virus RNA in the supernatant, viral RNA was isolated from 100 µL of supernatant by using a viral RNA isolation kit (Macherey Nagel, Dueren, Germany), and a first strand synthesis kit with ZIKV specific primers was used to synthesize cDNA (Thermo Fisher, Waltham, MA, USA). To perform real-time qPCR, synthesized cDNA was used as a template in a mixture of SYBR master mix (KAPA Biosystems, Switzerland) and ZIKV *NS5* specific primers (Forward: GTACATGGACTACCTATCCACC, Reverse: CTGACTAGCAGGCCTGACAAC). The qPCR reaction was carried out with the StepOnePlus™ Real-Time PCR system (Thermo Fisher, Waltham, MA, USA). Obtained cycle threshold (Ct) values were converted to the actual copy number of ZIKV *NS5* genes by using a standard curve. 

To generate the ZIKV *NS5* gene standard curve, a 105 bp ZIKV *NS5* PCR amplicon was cloned into a pJET1.2 vector (Thermo Fisher, Waltham, MA, USA). Logarithms (base 10) of concentrations of copy numbers were prepared from 10^9^ to 10^1^. The qPCR reaction was performed and obtained Ct values were converted to a standard curve by using GraphPad Prism 7.0 software (GraphPad Software, San Diego, CA, USA). The detection limit was 1000 copies of the ZIKV *NS5* gene.

### 2.10. Time of Drug Addition Assay

The fluorescent intensity assay that quantifies the ZsGreen expression from the ZIKV-ZsGreen vector was used in the time-of-addition assay. Briefly, 10^4^ Vero B4 cells per well were seeded in a 96-well black plate with transparent bottoms 1 day prior to the experiment. To perform the assay with pre-treatment conditions (−2 to 0 h), the media in wells were replaced with 10 µM benzavir-2 in DMEM, containing 1% FBS for 2 h at 37 °C. Then, all the wells were infected with 0.2 MOI of rZIKV-ZsGreen for 2 h at 37 °C and replaced with fresh DMEM. In the 0−2 h wells, 10 µM benzavir-2 was added together with the virus. At 2 hpi, the virus was removed from all of the wells and 10 µM benzavir-2 was added every second hour in 2 h pulses (0 to 2 h, 2 to 4 h, 4 to 6 h, or 6 to 8 h). For the 8 h sample (0 to 8 h), the cells were treated with 10 µM benzavir-2, together with the virus, for 2 h, and the medium was replaced with 10 µM benzavir-2 in DMEM for an extra 6 h. After 24 h of infection, the cells were fixed with 4% paraformaldehyde and washed with PBS. The fixed cells were counterstained with 300 nM DAPI in PBS. The Trophos plate runner HD (Trophos, Roche Group, Basel, Switzerland) identified all individual ZsGreen-expressing cells and quantified the fluorescence intensity in each well. 

### 2.11. Focus-Forming Assay to Study Antiviral Compound Inhibiton

Monolayers of Vero B4 cells were infected with ZIKV, TBEV, WNV, JEV, YFV, or DENV2 with a multiplicity of infection (MOI): 0.1 in DMEM, containing either DMSO, 2.5 µM benzavir-2, or 50 µM ribavirin, for 1 h at 37 °C 5% CO2. After 1 h of infection, the inoculum was replaced with DMEM 1% penicillin-streptomycin and 2% FBS, supplemented with either DMSO, 2.5 µM benzavir-2, or 50 µM ribavirin. Supernatants were harvested at 24 hpi (TBEV and ZIKV) or 48 hpi (WNV, JEV, YFV, and DENV2), and titers were determined using focus-forming assay, as previously described [28,29]. In brief, viruses were diluted in 10-fold and used for infection of Vero B4 cells. Viral foci were detected using primary monoclonal mouse antibodies, diluted by 1:1000, and directed against TBEV (19/1493) [30], flavivirus E (WNV, JEV, DENV2-HB112 ATCC), or YFV (CRC1689 ATCC) [29,30,31]. After primary antibody incubation, secondary horseradish peroxidase-conjugated anti-mouse antibodies (1:2000, Thermo Fisher Scientific) were added, followed by TrueBlue peroxidase substrate (KPL, Gaithersburg, MD, USA).

### 2.12. Statistical Analysis

Means and standard deviations (SDs) were calculated with GraphPad Prism 7.0 software. All statistical analyses were performed using one-way analysis of variance (ANOVA) and Dunnett’s post-hoc analysis was used for multiple comparisons test by GraphPad Prism 7.0 software; *p* < 0.05 was considered statistically significant.

## 3. Results

### 3.1. Benzavir-2 Inhibited ZIKV Derived Reporter Gene Expression in Vero B4 Cells

To explore the anti-ZIKV activity of benzavir-2, we assessed the inhibition of ZsGreen expression after infection of the recombinant ZIKV (rZIKV-ZsGreen). In this recombinant infectious virus, the ZsGreen gene was fused to the sequence encoding foot and mouth disease virus (FMDV) 2A autoprotease and cloned in-between two copies of the sequence encoding ZIKV capsid protein (Figure 1B). The infectivity of the rZIKV-ZsGreen was similar compared to wild-type ZIKV, as previously reported [21].

The inhibitory effect of the compounds (Figure 1A) on rZIKV-ZsGreen expression was assessed in a dose-dependent manner (Figure 2 and Table 1). Benzavir-2 displayed the highest potency among the compounds, with an EC_50_ value of 0.8 ± 0.1 µM (EC_90_ of 1.2 ± 0.3 µM). The EC_50_ value of ribavirin was 45.5 ± 13.5 µM (EC_90_ of 114.5 ± 18.0 µM), while no antiviral activity of favipiravir was detected in the tested concentration range (highest concentration was 100 µM). Benzavir-1, the precursor to benzavir-2, displayed at a lower potency (EC_50_ of 3.0 ± 0.1 µM and EC_90_ of 3.3 ± 0.1 µM) compared to benzavir-2, while the two structurally-related analogs, 17d and 35f, showed no antiviral activity or reduced potency (EC_50_ of 6.1 ± 0.1 µM and EC_90_ of 14.9 ± 0.9 µM, respectively). The results for benzavir-1, benzavir-2, 17d, and 35f are in line with our previous observations with herpes simplex virus (HSV) [18], human adenovirus (HAdV) [17], and Rift Valley fever virus (RVFV) [19].

### 3.2. Assessment of Host Cell Toxicity

The host cell toxicity of benzavir-2 and the structural analogs have previously been assessed using other cell types than Vero B4 [16,18,19]. As previously reported, solubility issues above 125 µM in cell media have been observed for benzavir-2. Hence, we set the maximum concentration for toxicity assessment to 100 µM for all compounds. The Vero B4 cell toxicity was analyzed 24 h post-treatment using a resazurin-based toxicity test, which is a colorimetric assay that measures the metabolic activity of living cells [32]. All compounds displayed cell viability well above 50% at the highest concentration of 100 µM, thus, no CC_50_ value could be measured and no selectivity index (SI) could be presented (Table 1).

### 3.3. Benzavir-2 Inhibited Plaque Formation of Two Wild-Type ZIKV Lineages

To further confirm the antiviral activity of benzavir-2 and to exclude that the observed inhibition was specific against rZIKV-ZsGreen, we analyzed whether benzavir-2 inhibited wt ZIKV plaque formation. The formation of plaques corresponds to a full infection cycle, from entry to egress of newly-formed viral particles. Based on the dose-response activity against the rZIKV-ZsGreen vector (Figure 2 and Table 1), we conducted a plaque formation assay in a dose-dependent manner by using benzavir-2 (from 20 to 0.6 µM) and ribavirin (from 200 to 6.3 µM) to observe plaque forming inhibition against the more clinically relevant wt ZIKV Asian lineage (BeH819015, Brazil strain). After 4 days of ZIKV infection, the Vero B4 cells were fixed and plaques were visualized by crystal violet staining and then quantified. Benzavir-2 showed significant reduction of plaque formation at all applied concentrations, while ribavirin only showed inhibition of plaque formation at 200 µM (Figure 3A). To further investigate benzavir-2 inhibition against other wt ZIKV lineages, we treated the African lineage (MR766, Uganda strain) with benzavir-2 at 2.5 µM. It showed a significant reduction of plaque formation, 94% reduction for the African lineage and 88% reduction for Asian lineage (Figure 3). 

### 3.4. Benzavir-2 Inhibited the Production of Infectious Progeny Virus

The reduction of wt ZIKV plaque formation by benzavir-2 was further investigated by analyzing the production of infectious progeny particles in the supernatant of infected cells. The supernatants from cells infected with the two wt ZIKV lineages, Asian and African, treated with 2.5 µM benzavir-2 (see above), were collected and used to infect Vero B4 cell layers. Infection efficiency was determined by observing the CPE, plaque formation, and qRT-PCR (see schematic diagram in Figure 4A). On day 2 of the post-progeny virus transfer, cells infected with progeny virus from benzavir-2 treated cells, displayed minor CPE. At day 4, the cells were fixed and stained with crystal violet to visualize the infection and plaque formation. Cells infected with progeny virus from benzavir-2-treated cells displayed only a few visible plaques. (Figure 4C).

To further quantify the antiviral effect, we harvested the supernatant on day 4, post-progeny virus transfer, and the viral RNA from the supernatant was measured using qRT-PCR. The reduction of the virus RNA copy number in the supernatant from benzavir-2 treated cells was 10^2.7^-fold (African ZIKV) and 10^3.9^-fold (Brazilian ZIKV) (Figure 4D). To conclude, benzavir-2 at 2.5 µM efficiently inhibited the production of the infectious progeny virus.

### 3.5. Benzavir-2 Mainly Affected the Early Stage of the ZIKV Infectious Cycle

To obtain insight into the mode-of-action of benzavir-2, a time-of-addition experiment was performed. Vero B4 cells were infected with rZIKV-ZsGreen and benzavir-2 was added at different time points; pre-treatment (−2 to 0 h) for every second hour in 2 h pulses (0–2 h, 2–4 h, 4–6 h, or 6–8 h) and for 8 h (0–8 h). Based on the growth kinetics of rZIKV-ZsGreen and the sensitivity of the ZsGreen detection in the fluorescent intensity assay, the analysis of virus infection was performed 24 h post-infection, and a 2-h treatment scheme with benzavir-2 at a relatively high concentration (10 µM) was selected (Figure 5A). When benzavir-2 was added at 2–4 h post-infection (hpi), the inhibitory effect was most pronounced, while the binding and entry steps (0–2 h) were less affected. Addition of benzavir-2 at later time-points post-infection also showed less inhibition. Although pre-incubation of the cells with benzavir-2 prior to infection (−2 h) had an inhibitory effect; it was much lower compared to the addition of benzavir-2 in the early phase (2–4 h) of infection (Figure 5B).

### 3.6. Benzavir-2 Displayed Potent Antiviral Activity against Several Flaviviruses

Members of the flavivirus genus cause severe morbidity and mortality to mankind worldwide. Since benzavir-2 displayed potent activity against the rZIKV-ZsGreen vector and two wt ZIKV lineages, we assessed if other flaviviruses of medical importance were inhibited. Vero B4 cells were infected with six members of the flavivirus genus (ZIKV, TBEV, WNV, JEV, YFV, or DENV2) in the presence of benzavir-2 (2.5 µM) or ribavirin (50 µM). The supernatant was harvested 24 h or 48 h post-infection and titers were determined by focus-forming assay. Ribavirin showed an approximately ten-fold significant reduction for four of the flaviviruses, ZIKV, TBEV, YFV, and DENV2 (viral titer reduction: ZIKV = 1.0 × 10^1^; TBEV = 1.2 × 10^1^; YFV = 0.9 × 10^1^; and DENV2 = 1.1 × 10^1^), while no significant activity was detected against JEV and WNV (Figure 6). Strikingly, 2.5 µM of benzavir-2 treatment showed a ≈ 10^3^–10^5^-fold significant reduction of infection in all six flaviviruses analyzed (viral titer reduction in: ZIKV = 1.0 × 10^3^; TBEV = 7.5 × 10^3^; WNV = 2.2 × 10^4^; JEV = 7.5 × 10^3^; YFV = 1.5 × 10^3^; and DENV2 = 5.6 × 10^4^). These results confirmed the potent broad-spectrum activity of benzavir-2.

## 4. Discussion

Flavivirus infections are a major public health problem [33]. Annually, millions of people are infected and flaviviruses cause significant health, economic, and social burdens worldwide [34]. The virus is endemic in many countries but can cause devastating outbreaks. In 2015, there was a widespread ZIKV outbreak in the Americas associated with severe neurological disorders (i.e., microcephaly and Guillain–Barré syndrome). In Africa in 2016, YFV re-emergence caused 600 deaths [10]. No antiviral drugs are available to treat flavivirus infections. Hence, there is an urgent need for proper antiviral therapies in order to control forthcoming outbreaks.

We have previously discovered novel anti-adenoviral compounds, and a subsequent medicinal chemistry program resulted in benzavir-2 [17], a potent broad-spectrum antiviral compound. Benzavir-2 presents potent antiviral activity against DNA-viruses (HAdV, HSV-1, and HSV-2, with respective acyclovir-resistant clinical HSV isolates), and recently we reported that benzavir-2 is active in vitro against a negative-sense single stranded RNA-virus, RVFV [16,18,19]. Here, we describe the antiviral activity of benzavir-2 against positive-sense single-stranded RNA viruses from the flavivirus genus, together with the activity of three structural analogs to benzavir-2 and two commercial antiviral drugs with previously-reported anti-flavivirus activity [20,35,36]. 

The recombinant ZIKV with ZsGreen-insertion (rZIKV-ZsGreen, Brazilian strain isolate) was a valuable screening tool. The infection was easily quantified and the infectivity of the recombinant virus was very similar to wt ZIKV [21]. We used this recombinant virus for primary assessment of the anti-ZIKV activity of benzavir-2. The dose-response studies revealed a sub-micromolar EC_50_ value of 0.8 ± 0.1 µM for benzavir-2 (EC_90_ value of 1.2 ± 0.3 µM), whereas the two commercial antiviral drugs, ribavirin and favipiravir, that were included as positive controls, had very moderate to no antiviral activity, respectively. 

Ribavirin is a well-known nucleotide analog and its anti-flavivirus activity has previously been reported [20,35]. Ribavirin showed moderate anti-ZIKV activity in our assay (EC_50_ value of 45.5 ± 13.5 µM and EC_90_ value of 114.5 ± 18.0 µM), consistent with previous reports displaying a variable efficacy range of ribavirin against ZIKV (EC_50_ between 20 and 48 µM) [37,38,39]. Favipiravir failed to display anti-ZIKV activity in our assay. Lei Cai et al. have showed an EC_50_ value above 100 µM (EC_50_ was 110.9 ± 13.1 µM) for favipiravir, which was the highest compound concentration in our assay [40]. Thus, antiviral activity might have been observed with increased concentration of favipiravir. 

Three structural analogs to benzavir-2 (Figure 1A), benzavir-1, 17d, and 35f were included in this study to assess whether the activity profile against ZIKV corresponds to our previous findings with HAdV [17], HSV [18], and RVFV [19]. In these studies, benzavir-2 was the most potent compound with EC_50_ values in the range of 0.6–1.6 µM, followed by the less potent benzavir-1 and 35f. Compound 17d was less efficacious and accurate EC_50_ could not be determined for all viruses. 

Interestingly, benzavir-2 and benzavir-1 follows the sigmoidal dose-response curve, as previously reported [18,19]. This results in a very narrow concentration range between EC_50_ and EC_90_ values. However, the reason for this phenomenon is unknown, but may contribute to the understanding of the mode-of-action for this compound class.

To confirm the antiviral activity of benzavir-2, we shifted to the therapeutically relevant wt ZIKV and conducted a plaque formation assay in a dose-dependent manner, using benzavir-2 (from 20 to 0.6 µM) and ribavirin (from 200 to 6.3 µM) against the Brazilian ZIKV strain. Benzavir-2 showed a significant reduction of plaque formation at all applied concentrations, while ribavirin showed a significant inhibition at 200 µM (Figure 3A). To examine the plaque inhibition activity of benzavir-2 against another ZIKV lineage, we used the African ZIKV (African MR766, Uganda strain) [41,42]. Previously, the plaque formation of these strains was described and quantification of plaque formation is ideal when assessing antiviral compounds, since the endpoint read-out, plaque formation includes the full infection cycle [43]. Benzavir-2 showed a significant inhibition of plaque formation at 2.5 µM for both ZIKV lineages, 94% for the African, and 88% reduction for the Asian lineage.

To further assess the inhibition of plaque formation by benzavir-2, we quantified the production of the infectious progeny virus. Benzavir-2 strongly inhibited the production of the progeny ZIKV virus particles released to the cellular surroundings. From a drug development and physiological perspective, a pronounced inhibition of the progeny virus production is favorable, because the ultimate aim of an antiviral drug is to diminish the viral spread.

Flaviviruses share several common cellular factors in their infectious cycle that might specifically be targeted by benzavir-2 [1,13,44]. Our previous findings of the broad-spectrum antiviral activity for benzavir-2, inhibiting both DNA and RNA viruses, indicated that benzavir-2 could act on a host cell target(s). To elucidate where in the ZIKV infectious cycle this target(s) could be present, we performed a time-of-addition analysis. It revealed that benzavir-2 was most effective at the early time-points of the ZIKV infectious cycle and did not inhibit the binding or entry to the same extent. The most pronounced inhibition was found when benzavir-2 was present between 2 to 4 h after infection, which could indicate an effect on protein translation and protein maturation or early replication and assembly of replication vesicles in the ZIKV infectious cycle. Previously, we performed similar time-of-addition analysis of benzavir-2 on RVFV [19]. There, the most pronounced inhibition was also in the early time-points, more precise between 0 and 2 h after infection. This indicated that RVFV binding, endocytosis, primary transcription, or protein translation might be inhibited. The discrepancy of the timing of benzavir-2 activity between RVFV and ZIKV might be explained by the difference in kinetics between the two viruses. RVFV is known to have a more rapid infectious cycle compared to ZIKV, hence possibly utilizes the target(s) that benzavir-2 acts upon earlier [45,46]. It also indicated that benzavir-2 was not interfering with general endocytosis and infection and might not inhibit membrane rearrangements needed in the ER for flavivirus replication as the negative stranded RNA virus RVFV does not share this feature [47]. 

The lack of available antiviral drugs to treat flaviviruses made us assess the antiviral activity of benzavir-2 on additional medically-important members of the flavivirus genus. The antiviral effect of benzavir-2 and ribavirin was assessed by focus forming assay on six different wild-type flaviviruses (BSL3 classified: TBEV, WNV, JEV, YFV, DENV2, and BSL2+ classified: ZIKV as a reference). Strikingly, benzavir-2 at 2.5 µM displayed potent antiviral activity, reducing viral titers with three to five orders of magnitude against all tested flaviviruses.

To conclude, benzavir-2 is a potent antiviral compound with broad-spectrum activity that spans across both DNA and RNA viruses. Here, we described the antiviral activity of benzavir-2 against flaviviruses, where there is a great-unmet medical need of antiviral therapy. The elucidation of the precise cellular target and the mode-of-action for benzavir-2 and its efficacy in vivo is ongoing, which will contribute to the development of new effective antiviral drugs.

## Figures and Tables

**Figure 1 viruses-12-00351-f001:**
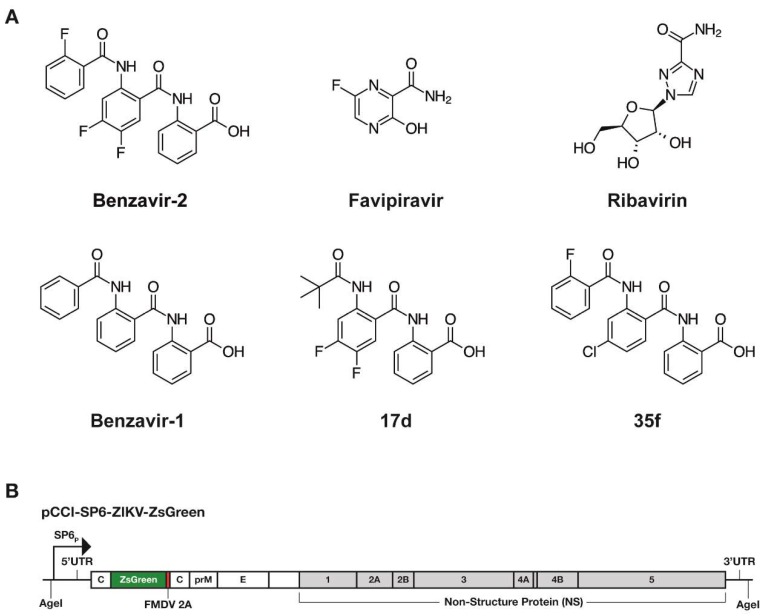
(**A**) Structures of the compounds used in this study. (**B**) Illustration of the recombinant Zika virus (ZIKV) containing the gene for *Zoanthus species* green fluorescence protein (rZIKV-ZsGreen). The virus was rescued from a plasmid containing a full-length infectious clone of a Brazilian ZIKV isolate ZsGreen (pCCI-SP6-ZIKV-ZsGreen). SP6p: SP6 phage promoter; foot and mouth disease virus (FMDV) 2A self-cleaving peptide.

**Figure 2 viruses-12-00351-f002:**
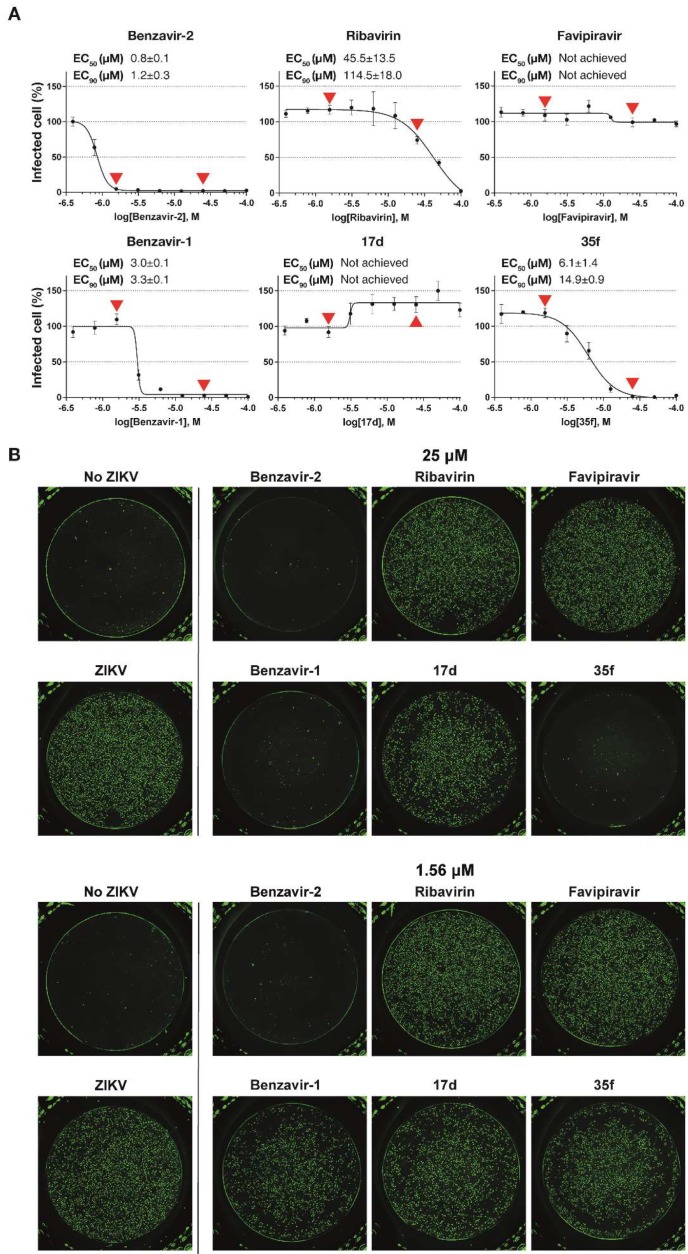
Dose-response curves and EC_50_/EC_90_ values of respective compounds against infection with the recombinant Zika virus (ZIKV) with the ZsGreen reporter (rZIKV-ZsGreen), measured by inhibition of ZsGreen expression. (**A**) Vero B4 cells were infected with rZIKV-ZsGreen (multiplicity of infection (MOI) = 0.2) in the presence of two-fold dilutions of the respective compound. After 24 h at 37 °C with 5% CO_2_, the virus/compound mixture was removed and the cells were washed and fixed with 4% paraformaldehyde. Individual ZsGreen-expressing cells were identified and quantified by Trophos plate runner HD®. From the quantified infection rate, the dose-response curve was generated and the EC_50_ value was calculated. Mean values and standard deviations from three independent experiments in triplicates are shown. (**B**) Representative picture of the fluorescent images of inhibition of rZIKV-ZsGreen infection at two selected concentrations (25 µM and 1.56 µM). The arrow heads (▲or▼) correspond to the selected concentrations of the respective compound.

**Figure 3 viruses-12-00351-f003:**
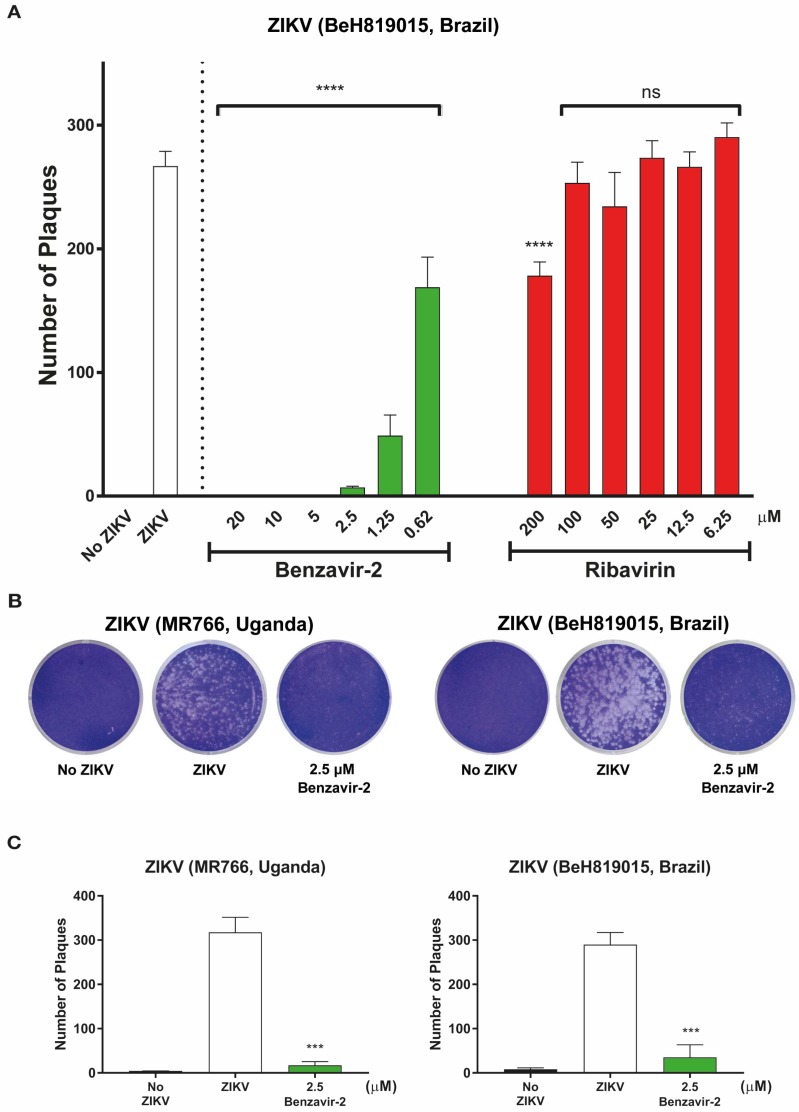
Antiviral activity of benzavir-2 and ribavirin against wild-type Zika virus (wt ZIKV) infection by plaque inhibition assay. Vero B4 cells were seeded in 12-well plates (2 × 10^5^ cells per well) and viruses (300 PFUs), together with benzavir-2 or ribavirin in Dulbecco’s modified Eagle medium (DMEM) with 1% carboxymethylcellulose (CMC), was added. The cells were incubated for 4 days. Next, cells were fixed with 4% paraformaldehyde and stained with 1% crystal violet solution. (**A**) Dose-dependent plaque-forming assay of the benzavir-2 effect against the Asian wt ZIKV lineage (BeH819015, Brazil strain). The assay was performed with 2-fold serial diluted benzavir-2 (from 20 to 0.62 µM) or 2-fold serial diluted ribavirin (200 to 6.25 µM). (**B**) Plaque forming assay at one benzavir-2 concentration (2.5 µM) for two wt ZIKV lineages (Asian BeH819015, Brazil strain and the African lineage MR766, Uganda strain). The results were visualized after crystal violet staining. (**C**). Quantification of the number of plaques in B. For all experiments, quantification data were analyzed by combining three independent experiments. Statistical significance was determined by one-way analysis of variance (ANOVA), followed by Dunnett’s multiple-comparisons test. *** *p* < 0.001, **** *p* < 0.0001; not significant (ns).

**Figure 4 viruses-12-00351-f004:**
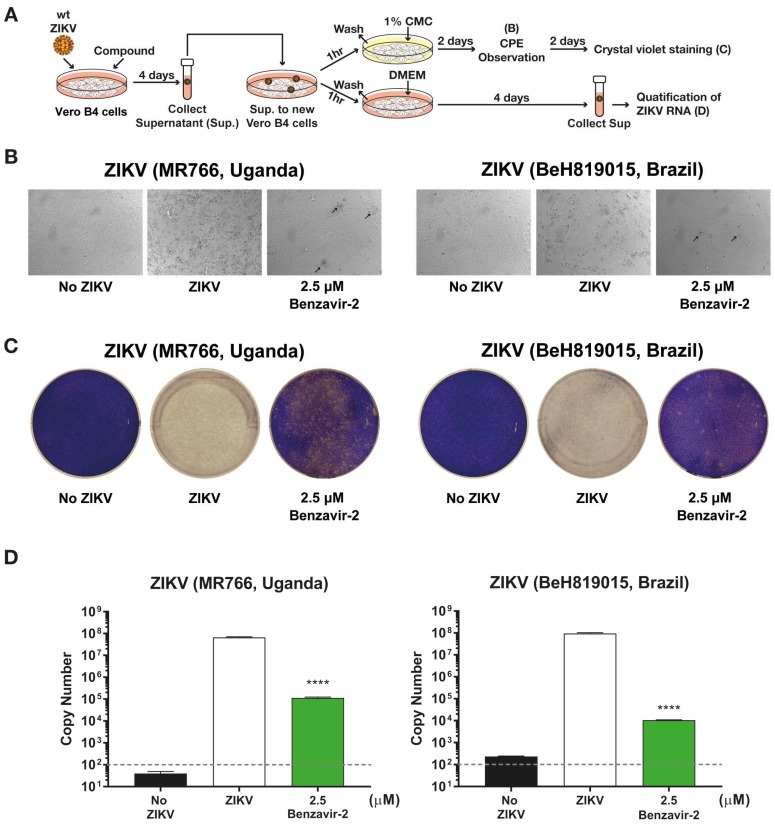
Inhibition of infectious progeny virus production by benzavir-2. (**A**) Schematic diagram for analysis of progeny virus production. Vero B4 cells were seeded in 12-well plates (2 × 10^5^ cells per well) and the virus (300 PFUs), together with 2.5 µM of benzavir-2. Cells were incubated for 4 days. The supernatants were collected from two different ZIKV lineage infected cells with or without compounds. Next, the collected supernatants were transferred to a new Vero B4 cell layer for 1 h and replaced with DMEM with 1% CMC (for cytopathic effect (CPE) observation and crystal violet staining) or with DMEM (for qRT-PCR), respectively. (**B**) Microscopy images of the virus-infected cells on day 2. Black arrows indicate CPE onset. (**C**) Plaque formation determined by crystal violet staining on day 4. Representative images are shown in B and C. (**D**) Quantification ZIKV RNA from collected supernatant on day 4 post-infection. The grey-dashed line represents the detection limit of qRT-PCR. Mean values and standard deviations are shown from three independent experiments. The statistical significance was determined by one-way ANOVA, followed by Dunnett’s multiple-comparisons test. **** *p* < 0.0001.

**Figure 5 viruses-12-00351-f005:**
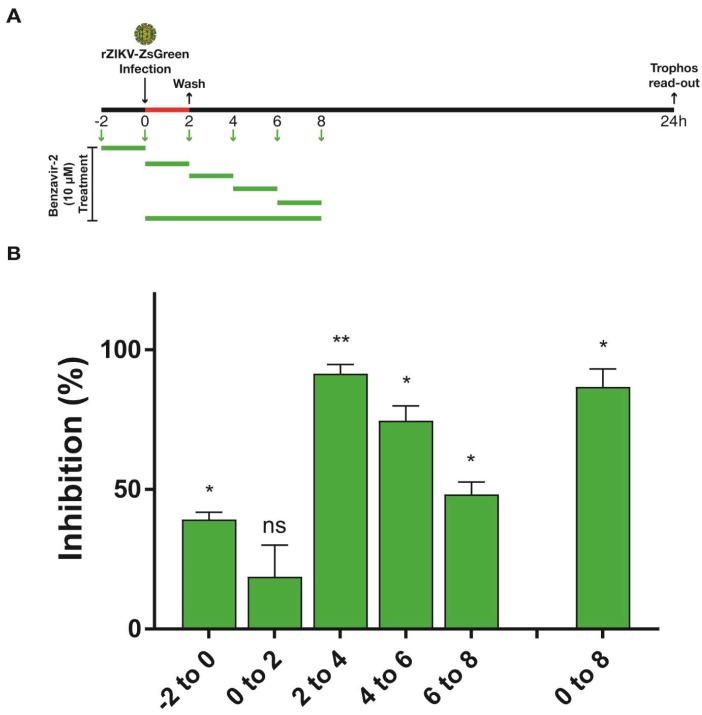
Time-of-addition assay of benzavir-2 using rZIKV-ZsGreen. 10^4^ Vero B4 cells per well were seeded in a 96-well black plate with transparent bottoms the day prior to the experiment. (**A**) Illustration displaying the experimental set-up. Briefly, to perform the assay with the pre-treatment condition (−2–0 h), the media from each well was replaced with 10 µM benzavir-2 in DMEM, containing 1% fetal bovine serum (FBS) for 2 h at 37 °C. Thereafter, cells were infected with 0.2 MOI of rZIKV-ZsGreen for 2 h at 37 °C, followed by the addition of fresh DMEM. In the 0–2 h wells, 10 µM benzavir-2 was added together with the virus for 2 h and was replaced with fresh DMEM. For the 2 h benzavir-2 treatment intervals, the virus was removed from all the wells 2 h post-infection (hpi) and was replaced with 10 µM benzavir-2 in DMEM every second hour (2–4 h, 4–6 h, or 6–8 h). To treat benzavir-2 for 8 h (0–8 h), the cells were treated with 10 µM benzavir-2 together with the virus for 2 h and the medium was replaced with 10 µM benzavir-2 in DMEM for an extra 6 h. (**B**) After 24 h of infection, the Trophos plate runner HD identified all individual ZsGreen-expressing cells and quantified the fluorescence intensity in each well. Mean values and standard deviations from three independent experiments are shown. The statistical significance was determined by one-way ANOVA, followed by Dunnett’s multiple-comparisons test. * *p* < 0.05, ** *p* < 0.01; not significant (ns).

**Figure 6 viruses-12-00351-f006:**
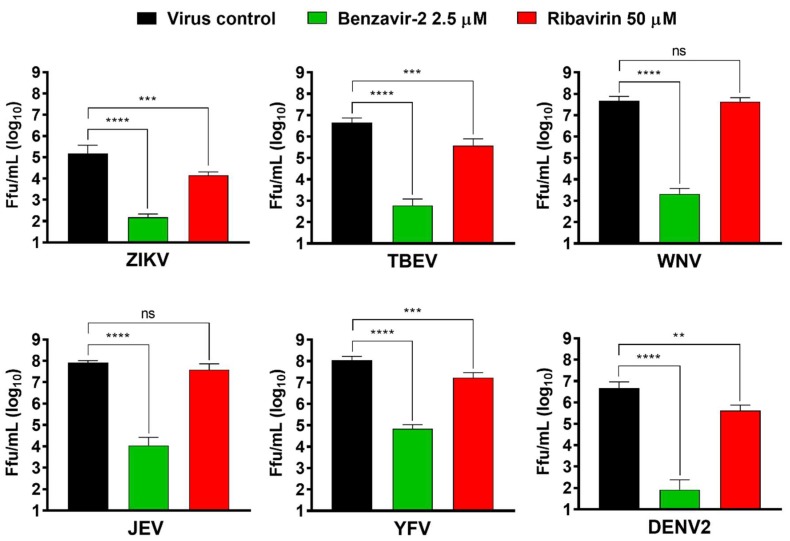
Benzavir-2 inhibition of flavivirus infection analyzed with a focus-forming assay. Monolayers of Vero B4 cells were infected with Zika virus (ZIKV), tick-borne encephalitis virus (TBEV), West Nile fever virus (WNV), Japanese encephalitis virus (JEV), yellow fever virus (YFV), or dengue virus type 2 (DENV2) with a multiplicity of infection (MOI) of 0.1 in the presence of benzavir-2 (2.5 µM), ribavirin (50 µM), or DMSO in the virus control. After 1 h, virus inoculum was removed, the cells were washed, and media containing the compounds or DMSO was added. Supernatants were harvested at 24 h post-infection (hpi) (TBEV and ZIKV) or 48 hpi (WNV, JEV, YFV, and DENV2), and virus titers were measured by focus-forming assay. Mean values and standard deviations from two independent experiments performed in triplicates are shown. The statistical significance was determined by one-way ANOVA, followed by Dunnett’s multiple-comparisons test. ** *p* < 0.01, *** *p* < 0.001, **** *p* < 0.0001; not significant (ns).

**Table 1 viruses-12-00351-t001:** Inhibition efficacy (EC_50_ and EC_90_) of compounds used in this study against rZIKV-ZsGreen and Vero B4 cell toxicity.

	Inhibition Efficacy (µM)		Vero B4 Cell Viability (%)		
	EC_50_	EC_90_	100 µM	50 µM	25 µM
**Benzavir-2**	0.8 ± 0.1	1.2 ± 0.3	94.3 ± 1.8	96.8 ± 0.6	100.5 ± 1.8
**Ribavirin**	45.5 ± 13.5	114.5 ± 18.0	93.5 ± 3.4	95.0 ± 2.8	96.0 ± 1.0
**Favipiravir**	Not achieved	Not achieved	97.9 ± 2.2	95.0 ± 1.4	97.4 ± 0.7
**Benzavir-1**	3.0 ± 0.1	3.3 ± 0.1	88.1 ± 2.1	88.3 ± 1.4	98.8 ± 3.4
**17d**	Not achieved	Not achieved	104.9 ± 0.9	106.1 ± 4.2	105.1 ± 0.1
**35f**	6.1 ± 1.4	14.9 ± 0.9	92.1 ± 3.9	96.5 ± 3.7	104.8 ± 5.0

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
