# Peer review of "Antiviral Activity of Benzavir-2 against Emerging Flaviviruses"

_viruses, 2020, doi:10.3390/v12030351_

Round 1
Reviewer 1 Report
This manuscript by Gwon et al., describes efficacy testing of benzavir-2 in preventing propagation of Zika virus (ZIKV) in cell culture assays. These assays initially tested cells treated with different concentrations of benzavir-2 two hours prior to virus infection. Subsequent assays evaluated post-infection treatment approaches and tested the compound against other flaviviruses. These studies found that 2.5 uM benzavir-2 was effective at reducing ZIKV propagation using a pre-treatment approach, but that repeated dosing post-infection was more effective. Further, 2.5 uM benzavir-2 was shown to be effective at limiting propagation of multiple flaviviruses when treatment was performed concurrent with viral infection and maintained for 24h post-infection.
General comments: Initial testing of benzavir-2 at multiple concentrations demonstrated a marked antiviral effect up to 25 uM with little or no cytotoxicity. It isn’t clear to me why continued testing was performed using a 2.5 uM dose (~2x EC90) rather than a 5 uM dose that was shown to completely inhibit virus propagation in plaque assay tests (Fig 3).
Answer: We performed the dose-response first and found that when we used the fluorescence-intensity assay, EC50 was 0.8µM and EC90 was 1.2µM for benzavir-2 (see table 1). We then performed a dose-response using the plaque-forming assay, and found that the EC50 and EC90 was very similar (see fig 3A).
In the fluorescence-intensity assay, 2.5µM of benzavir-2 corresponded to 99.953% inhibition, and in the plaque assay, 2.5µM corresponded to 97.37% inhibition of plaque formation. Thus, we decided to use 2.5µM because it was the lowest concentration of benzavir-2 that resulted in basically complete inhibition.
Further, I didn’t understand the rationale for analyzing virus progeny production. This experiment does show that benzavir-2 reduces viral load at a 2.5 uM treatment dose, but the fact that virus propagates once treatment is effectively removed is underwhelming. Personally, I would have left this experiment out or done the test using a 5 uM treatment dose as that appeared to have a higher degree of virus inhibition based on your plaque assay data.
Answer: Thank you for the question. To show that virus progeny production is inhibited is an assay that is usually implemented in most antiviral studies, and something that most reviewers ask for if it is not performed. The rationale for using several orthogonal assays in our study (fluorescence intensity, progeny production, RNA quantification) was to verify that benzavir-2 was genuinely active against the virus, and not only perform one assay with potential biases. Regarding 5µM instead of 2.5µM, please see our answer to the first question. 2.5µM was 97.37% inhibition and we believe that it was sufficient to prove inhibition. The reviewer also writes: “the fact that virus propagates once treatment is effectively removed is underwhelming.” Regarding the results in section 3.3, the propagation after removal of treatment was not attempted or shown here. If the reviewer meant the following time-of-addition experiment, please also see the answer below (question 6).
Specific comments:
Section 2.2: Please provide the source of your Vero B4 cells particularly since this clonal population is not in common use.
Answer: The source of the Vero B4 cell line is:
Meyer B, García-Bocanegra I, Wernery U, Wernery R, Sieberg A, Müller MA, Drexler JF, Drosten C, Eckerle I. Serologic assessment of possibility for MERS-CoV infection in equids. Emerg Infect Dis. 2015;21:181–182.
It has been inserted into the manuscript (line 75) according to the reviewer’s suggestion.
Line 73: synthesize
Answer: The manuscript has been amended as suggested, “synthesis” has been removed and replaced with “synthesize”
Line 139: Meant to be ‘neutral-buffered formalin’?
Answer: The manuscript has been amended as suggested, “4% paraformaldehyde in neutral buffer” has been removed and replaced with “neutral-buffered formalin (4%)”
Figure 5: Were the treatment regimens described in figure 5, specifically 2-4, 4-6 and 0-8 treatment regimens also assayed in your post-treatment cell culture passage assay (figure 3A)? It would be useful to know if there was marked virus breakthrough after these treatments.
Answer: Thank you for the suggestion. No, the treatment regimens were not assayed in a progeny production assay. The progeny production assay was performed with wild-type ZIKV after 4 days of infection, but in the time-of-addition experiments we used the fluorescence intensity assay to be able to study early events of virus transcription, translation and expression of the reporter gene up to only 8 hours. We believe that for these short time-periods there will not be enough progeny virus produced, if any even for the control.
REVIEWER 2:
The authors present in this manuscript the activity of benzavir-2 drug on Zika virus (ZIKV). By using ZsGreen expressing vector, they show the inibitory effect of benzavir-2 on ZIKV and also on others flaviviruses.
The experiments are clearly presented and the inhibitory effect of benzavir-2 is convincing.
Reviewer’s question no. 1
Minor remark on title of paragraph 3.5: benzavir-2 mainly effected the early stage of the ZIKV infectious cycle. effected could be replaced by affected
Answer: “effected” has been amended to “affected” according to the Reviewer’s suggestion (line 299).